# Predicting Immunotherapy Outcomes in Older Patients with Solid Tumors Using the LIPI Score

**DOI:** 10.3390/cancers14205078

**Published:** 2022-10-17

**Authors:** Monica Pierro, Capucine Baldini, Edouard Auclin, Hélène Vincent, Andreea Varga, Patricia Martin Romano, Perrine Vuagnat, Benjamin Besse, David Planchard, Antoine Hollebecque, Stéphane Champiat, Aurélien Marabelle, Jean-Marie Michot, Christophe Massard, Laura Mezquita

**Affiliations:** 1Department of Medical Oncology, Gustave Roussy Cancer Campus, 94850 Villejuif, France; 2Faculty of Medicine, Sorbonne University, 75006 Paris, France; 3Drug Development Department, Gustave Roussy Cancer Campus, 94850 Villejuif, France; 4Oncology, Hôpital Européen George Pompidou, 75015 Paris, France; 5Senior Unit, Department of Cancer Medecine, Gustave Roussy Cancer Campus, 94850 Villejuif, France; 6Laboratory of Translational Genomics and Targeted Therapeutics in Solid Tumors, Medical Oncology Departement, Hospital Clinic, Institut d’Investigacions Biomèdiques August Pi Sunyer (IDIBAPS), 08036 Barcelona, Spain

**Keywords:** LIPI score, immune checkpoint inhibitors, older patients, neutrophils, aging

## Abstract

**Simple Summary:**

The Lung Immune Prognostic Index (LIPI) is a score that combines pretreatment dNLR (neutrophils/(leukocytes − neutrophils) and lactate dehydrogenase (LDH) and is correlated with outcomes in patients with non-small-cell lung cancer treated with anti PD-(L)1 but has not been validated in an older cohort of patients. LIPI is associated with poorer overall survival in older patients. LIPI is a simple and accessible worldwide tool that could serve as a prognostic factor and can be useful in identifying patients who will not benefit from such treatment.

**Abstract:**

Immunotherapy with immune checkpoint blockers (ICB) represents a valid therapeutic option in older patients for several solid cancer types. However, most of the data concerning efficacy and adverse events of ICB available are derived from younger and fitter patients. Reliable biomarkers are needed to better select the population that will benefit from ICB especially in older patients who may be at a higher risk of developing immune-related adverse events (irAEs) with a greater impact on their quality of life. The Lung Immune Prognostic Index (LIPI) is a score that combines pretreatment dNLR (neutrophils/[leukocytes − neutrophils]) and lactate dehydrogenase (LDH) and is correlated with outcomes in patients treated with ICB in non-small-cell lung cancer. We aimed to assess the impact of LIPI in ICB outcomes in a dedicated cohort of older patients. The primary objective was to study the prognostic role of LIPI score in patients aged 70 years or above in a real-life population treated with anti-programmed death-(ligand)1 (anti PD-(L)1). dNLR and LDH were collected in a prospective cohort of patients aged 70 years or above treated with PD-(L)1 inhibitors with metastatic disease between June 2014 and October 2017 at Gustave Roussy. LIPI categorizes the population into three different prognostic groups: good (dNLR ≤ 3 and LDH ≤ ULN—upper normal limit), intermediate (dNLR > 3 or LDH > ULN), and poor (dNLR > 3 and LDH > ULN). Anti PD-(L)1 benefit was analyzed according to overall survival (OS), progression free survival (PFS), and overall response rate (ORR) using RECIST v1.1. criteria. In the 191 older patients treated, most of them (95%) were ICB-naïve, and 160 (84%) had an ECOG performance status of 0–1 with a median age at ICB treatment of 77 (range, 70–93). The most common tumor types were melanoma (66%) and non-small-cell lung cancer (15%). The median follow-up duration was 18.8 months (95% CI 14.7–24.2). LIPI classified the population into three different groups: 38 (23%) patients had a good LIPI score, 84 (51%) had an intermediate LIPI score, and 43 (26%) had a poor LIPI score. The median OS was 20.7 months [95% CI, 12.6–not reached] compared to 11.2 months [95% CI, 8.41–22.2] and 4.7 months [95% CI, 2.2–11.3] in patients with a good, intermediate, and poor LIPI score, respectively (*p* = 0.0003). The median PFS was 9.2 months [95% CI, 6.2–18.1] in the good LIPI group, 7.2 months [95% CI, 5.4–13] in the intermediate LIPI group, and 3.9 months [95% CI, 2.3–8.2] in the poor LIPI group (*p* = 0.09). The rate of early death (OS < 3 months) was 37% in the poor LIPI group compared to 5% in the good LIPI group (<0.001). Poor LIPI score was associated with a poorer outcome in older patients treated with anti PD-(L)1. LIPI is a simple and accessible worldwide tool that can serve as a prognostic factor and can be useful for stratification benefit from ICB.

## 1. Introduction

Immune checkpoint blockers (ICBs) have drastically changed the therapeutic landscape and the prognosis of cancer patients [1,2,3], becoming the standard of care in a wide spectrum of solid tumors [4,5]. ICBs are an attractive therapeutic option in older patients with fewer toxicities compared to cytotoxic chemotherapies [6], but it is unclear whether age-related changes in the immune system called immunosenescence might negatively influence antitumor response and consequently affect the efficacy and safety of these drugs [7]. Few dedicated studies have explored ICBs specifically in this population. Data from clinical trials usually include a small number of older patients and do not represent the complexity and heterogeneity of this population. In these pivotal trials, older patients are usually fit and selected in accordance with very stringent criteria. Consequently, few real-life data are available on the efficacy and toxicity of immunotherapy in older patients. To date, there are no significant differences in terms of efficacy, whereas increased toxicity is discussed [6]. In contrast, older age is one of the clinical factors associated with the phenomenon of hyperprogressive disease [8]. Identifying the patients who will experience this aggressive pattern during ICB therapy is particularly relevant in a vulnerable or frail population, such as the older population. Predictive biomarkers have already been described in several tumor types such as programmed death ligand 1 (PD-L1) expression [9,10], tumor mutational burden (TMB), or mismatch repair deficiency (dMMR) [11,12]. However, a combined score of clinical and biological parameters is still relevant in everyday clinical practice. The Lung Immune Prognostic Index (LIPI) has demonstrated a strong correlation with immunotherapy outcomes, first in non-smal—cell lung cancer, but also in other solid tumor types [13]. This is a simple and accessible tool based on two blood parameters at baseline: the derived neutrophil-to-lymphocyte ratio (dNLR) (neutrophils/[leukocytes − neutrophils]) > 3 and the lactate dehydrogenase (LDH) levels > upper limit of normality (ULN) [14]. LIPI could be an interesting tool in older patients eligible for ICBs. Although all the cohorts exploring LIPI enrolled patients with >70 years old, the clinical impact of LIPI in this specific older population has not been formally studied, and it remains unknown in a real-life setting. We aimed to assess the impact of LIPI on ICB outcomes in a large real-life cohort of older patients with advanced solid tumors treated with anti PD-(L)1.

## 2. Materials and Methods

### 2.1. Patients

We conducted a single-center study of a cohort of 603 patients with advanced disease treated with PD-(L)1 inhibitors in a real-life setting, i.e., following marketing authorization, as a part of patient early access programs for unapproved indications or as compassionate use between June 2014 and October 2017 at Gustave Roussy and registered in the prospective pharmacovigilance database REISAMIC (“Registre des Effets Indésirables Sévères desAnticorps Monoclonaux Immunomodulateurs en Cancérologie”) [15]. Among them, we considered eligible for our study the cohort of patients aged ≥70 years old. Patient characteristics and biological data at baseline (e.g., complete blood cell counts, LDH, and albumin) were collected. Radiological responses were performed using irRECIST criteria [16].

### 2.2. LIPI Score

The LIPI score was calculated on the basis of the dNLR (neutrophils/[leukocytes − neutrophils] > 3) and LDH (>ULN). Cutoff values of dNLR and LDH were chosen according to the results of previous reports [14]. LIPI categorizes the population into three different prognostics groups: good (dNLR ≤ 3 and LDH ≤ ULN), intermediate (dNLR > 3 or LDH > ULN), and poor (dNLR > 3 and LDH > ULN). This study was approved by the Institutional Review Board of Gustave Roussy (Commission Scientifique des Essais Thérapeutiques). 

### 2.3. Statistical Analysis

Median (IQR) values and proportions (percentage) were used for continuous and categorical variables, respectively. Median and proportions were compared using the Wilcoxon–Mann–Whitney test and the chi^2^ test (or the Fisher’s exact test, if appropriate), respectively. Progression-free survival (PFS) was calculated from the date of first immunotherapy administration until disease progression or death due to any cause, whichever occurred first. Overall survival (OS) was calculated from the date of the first immunotherapy administration until death due to any cause. Early death rate included all cases of death within 12 weeks of anti PD-(L)1 therapy. Survival analyses were performed using the Kaplan–Meier method and the log-rank test. Follow-up was calculated using the reverse Kaplan–Meier method. The association of demographic, clinical, and biological factors with survival was assessed with univariate and multivariate Cox proportional-hazard models, providing a hazard ratio (HR) and its 95% confidence interval (CI). Variables included in the final multivariate model were selected according to their clinical relevance and statistical significance in univariate analysis (*p*-value cutoff = 0.10). Predictive factors of disease control were tested with logistic regression in univariate and multivariate analyses. All analyses were performed using R software version 4.1.0 (accessed on 18 May 2021). A *p*-value <0.05 was considered statistically significant, and all tests were two-sided.

## 3. Results

### 3.1. Patient Characteristics

A total of 191 older patients (≥70 years old) were enrolled in the analysis (Figure 1). Baseline characteristics of the population are summarized in Table 1. Median age was 77 years old (range 70–93), and the median follow-up was 18.8 months. Among them, 74 were female (39%) and 117 were male (61%). The most frequent solid tumor types were melanoma (66.5%, 127 patients), non-small-cell lung carcinoma (NSCLC) (15%, 29 patients), and small-cell lung cancer (SCLC) (11.5%, 22 patients). The median ECOG performance status (PS) of the whole population was 1 (0–4). Median previous line of treatment was 1.03. All patients enrolled in this study received ICBs as monotherapy. Patients were mostly treated with pembrolizumab (114 patients, 59.6%), nivolumab (68 patients, 35.6%), atezolizumab (seven patients, 3.6%), and avelumab (two patients, 1%). 

### 3.2. Pretreatment LIPI Score 

At baseline, dNLR was available in 182 patients (95%). The median dNLR was 6.04 (0–48.7), and dNLR was >3 in 128 patients (70.3%). High dNLR was associated with poor PFS [HR 1.50; 95% CI 1.02–2.19; *p* = 0.04] and poor OS [HR 2.10; 95% CI, 1.34–3.27); *p* = 0.001]. At baseline, LDH was available in 166 patients (86.9%). The median of LDH was 300.43 UI/L (119–1965)], and LDH was high in 60 patients (36.1%) patients. LDH > ULN was not significantly associated with poor PFS [HR 1.29; 95% CI 0.89–1.87; *p* = 0.18] but was associated with poor OS [HR 1.69; 95% CI, 1.14–2.53. *p* = 0.01]. In the overall population, LIPI was evaluable in 164 patients (85.9%). Among them, LIPI classified the older population into three different groups: 38 patients (23%) had a good LIPI score, 83 (51%) had an intermediate LIPI score, and 43 (26%) had a poor LIPI score.

### 3.3. Clinical Profile According to the LIPI Score

The baseline characteristics according to the LIPI groups are summarized in Table 1. All the patients with a good LIPI score had an ECOG PS 0–1 before immunotherapy (38 patients, 100%) vs. 73 patients (87.9%) with an intermediate LIPI and 10 patients (12%) with poor LIPI. The LIPI poor was most likely associated with ECOG PS ≥ 2 (25.5%).

### 3.4. LIPI Is Correlated with ICB Outcomes

With a median follow-up of 18.8 months [95% CI 14.7–24.2], the median PFS and OS for the whole population were 6.2 months [95% CI, 5.4–8.3] and 11.2 months [95% CI, 8.4–18.4], respectively. LIPI was significantly correlated with OS (*p* = 0.0003) but not with PFS (*p* = 0.09). According to the LIPI groups, the median OS was 20.7 months [95% CI, 12.55-NR] in the good LIPI group compared to 11.2 months [95% CI, 8.41–22.2] in the intermediate LIPI group and 4.7 months [95% CI, 2.2–11.3] in the poor LIPI group (*p* = 0.0003) (Figure 2A). The median PFS was 9.2 months [95% CI, 6.2–18.1] in patients with a good LIPI score compared to 7.2 months [95% CI, 5.4–12.9] in the intermediate and 3.8 months [95% CI, 2.3–8.2] in the poor LIPI score group (*p* = 0.09) (Figure 2B). The multivariate analysis included gender, histology, site of metastasis, performance status, albumin, and LIPI score. Regarding gender and histology, we observed that the LIPI and LDH were identically distributed among the different histologies. Interestingly, patients with other histologies had lower dNLR compared with the other patients. The same observation was made with male patients (higher dNLR compared with women, but same repartition in the LIPI groups). In order to take into account these potential confounding factors, we included them in the multivariate analysis, showing that LIPI score was an independent factor for OS [HR of 2.77 for the poor group; *p* < 0.008] but not for PFS [HR of 1.22 for the poor group; *p* = 0.36] (Table 2). In the multivariable Cox models, moreover, corticosteroid use and LIPI were independent prognostic factors for OS. We further investigated the prognostic value of LIPI in patients with or without corticosteroid use and found no difference in term of prognostication (Appendix A).

### 3.5. LIPI Is Correlated with ICB Response

In addition, we studied the impact of LIPI in the ICB response. Overall, the ORR was 31.7% and DCR 34.1%. No significant differences were observed in terms of ORR or DCR. According to the LIPI groups, the ORR was 44% in the good LIPI group, 33% in the intermediate LIPI group, and 26.8% in the poor LIPI group (*p* = 0.2) (Figure 3). The disease control rate (DCR) was 49% in the good LIPI group, 38% in the intermediate group, and 27% in the poor group (*p* = 0.14).

### 3.6. Early Death and Response Rate

Lastly, we studied whether LIPI was correlated with early death, as one of the most relevant aggressive patterns of ICB failure. In the overall population, the early death rate was 18%. LIPI was significantly associated with early death (*p* < 0.001). The rate of early death was 37% in the poor LIPI group vs. only 5% in the good LIPI group.

### 3.7. Pretreatment LIPI Score and Immune-Related Events (irAEs)

In the LIPI evaluable population, a total of 63 patients (33%) presented irAEs ≥ grade 2 (Table 3). Most of them were skin toxicities followed by thyroid, gastro-intestinal, and liver toxicities. By LIPI groups, we observed a higher rate of irAEs in the good LIPI group, with 17 events (45%) vs. 26 in the intermediate LIPI group (31%) and 13 (30%) in the poor LIPI group; however, there were no significant differences (*p* = 0.276).

## 4. Discussion

Here, we reported, for the first time, the impact of the LIPI as a prognostic biomarker in a real-life population of older patients treated with immunotherapy. In our study, we observed that, in the older population, 26% of the population with a poor LIPI score had significantly poorer outcomes (median OS of 4.7 months; 37% of early deaths) compared to intermediate and good LIPI groups. In addition, in our cohort of 191 patients, LIPI was an independent factor for OS [HR of 2.77 for the poor group; *p* < 0.008], providing relevant prognostic impact in this specific older population. Our data are consistent with previous LIPI studies. LIPI was initially studied in a cohort of pretreated patients with NSCLC, with 36% good, 49% intermediate, and 15% poor groups [14]. Interestingly, the distribution of LIPI groups reported later in other cohorts with other tumor types was quite similar, with the rate of LIPI poor between 14.5% and 6.3% [13,14,17]. In our study, the distribution was 26%, 51%, and 23% for poor, intermediate, and good LIPI, consistent with previous data reported, raising the hypothesis of LIPI as a pan-tumor biomarker for immunotherapy, most likely related to the immune context of the patient and not the tumor type [18,19]. However, in our cohort of older patients, PFS was not influenced by the baseline LIPI score (*p* = 0.09). We hypothesize that the lack of significant correlation between LIPI score and PFS can be related to the presence of different tumor types in our cohort with a different response rate and duration of response to anti PD-(L)1. The most frequent tumor type was indeed melanoma, which is not commonly associated with a highly inflammatory state such as NSCLC. This could explain the weak correlation among LIPI score and PFS. It is known that a high blood level of circulating neutrophils is a negative prognostic factor in patients with cancer [20]. Therefore, the neutrophil-to-lymphocyte ratio (NLR) and the derived neutrophil-to-lymphocyte ratio (dNLR) have been used to develop clinical indicators of systemic inflammation [13]. High dNLR has been associated with shorter survival also in patients with various tumor types, including pancreas, bladder, and renal cancer [13]. Older age is associated with a low-grade inflammatory systemic environment, known as inflammaging. In the aging population, neutrophil numbers do not change [21], and their adhesion to the endothelium appears unaltered, but a recent study evidenced that neutrophils from the sixth decade of life have a decline in the capacity of their migration and in the phagocytic ability for opsonized bacteria [22]. How neutrophils change to respond to environmental challenges is yet unclear, and it is unknown whether it is possible to talk of senescent neutrophils as it is recognized in T cells and monocytes/macrophages [23]. Franceschi et al. hypothesized that the low-grade inflammatory systemic environment seen with age (inflammaging) may generate epigenetic changes in cells such as DNA methylation [24,25], which may impact cellular phenotypes and functions [26], resulting in an altered response on immune cells to inflammatory stimuli [27] that could explain different responses to ICB. Mesquita et al. proposed LIPI as a potential predictive factor in the first cohort reported on the basis of a lack of prognostic impact in a cohort exclusively treated with chemotherapy [14]. However, in other works published later, LIPI was also associated with patient outcomes regardless of treatment therapy. Nevertheless, additional works have shown a higher potential to stratify the magnitude of benefit from immunotherapy than with other therapies [13,28]. In our work, we only assessed the prognostic role of LIPI, with no control cohort to explore this interesting hypothesis. In addition, although we observed a higher rate of responses (44% ORR, 49% DCR) in the LIPI good group, no statistical differences were observed.

Over the last few years, a new aggressive pattern of progression under immunotherapy has been reported [29], including hyperprogressive disease (HPD), fast progressors (FP), and early death (ED). There is still no consensus on the definition of these phenomena [30], but all of them are strongly associated with poor OS. In a previous work, LIPI was correlated with HPD according to the definition of Kim et al. [31], but this association has not been consistently observed with other HPD definitions. In our work, we identified a strong correlation between LIPI poor and ED, with 37% of cases in this group. These data are in line with a work very recently reported with a score based on dNLR (at baseline and at second cycle) [32]. Persistently high dNLR at both timepoints was correlated with early ICB failure and ED, as in our study.

These data are relevant in suggesting that LIPI score could be a real prognostic tool for clinicians helping to select the patients who will not benefit from an ICB treatment and, most importantly, who are at risk of early death. Even though there are limited data, these early results suggest that pretreatment LIPI score could also help identify older patients at higher risk of developing this aggressive pattern of progression with ICB, confirming its prognostic role. However, LIPI should not be considered as a unique predictive marker able to identify immunotherapy as the appropriate treatment. It reflects the proinflammatory status of the patient, and it is strongly correlated with immunotherapy survival; however, other additional factors with a predictive value already demonstrated such as PD-L1, TMB, and MSI are to be considered [2,10,11]. LIPI is a host-related biomarker that should be considered integrated, together with other tumor-related biomarkers, in order to reflect the complexity of the immune system.

Interestingly, our study showed a correlation between good LIPI score and irAEs. We hypothesize that patients with a good LIPI score are the group of patients with “favorable responses” to immunotherapy, which has been reported as associated with the development of irAEs [6]. In addition, the LIPI good group is also treated for a longer period of time compared to patients with a poor and intermediate LIPI; thus, the exposure to IBC and risk of irAEs is greater. Integrating LIPI in the comprehensive geriatric assessment (CGA) could help clinicians in the decision-making process, evaluating the estimated benefit of ICB and the risk for irAEs. Gomes et al. [33] investigated for the first time the role of geriatric assessment for older patients treated with ICB, showing that the G8 score [34] (a screening tool developed for cancer patients aged 70 years and consisting of eight questions) was able to identify vulnerable and frail older patients with a higher risk of hospital admission and a higher risk of death. However, to our knowledge, there are no biological parameters or indices such as LIPI integrated in these geriatric scores to date. Our study had several limitations mostly derived from the data collection, including missing clinical and pathological data such as the PD-L1 status and other causes of death (cancer-related or non-cancer-related). In a geriatric population, other comorbidities could indeed have a consistent impact as non-cancer-related causes of death. Moreover, the response assessment was not homogeneously performed. Furthermore, in the study, different tumor types were included with different types of response to anti PD(L)-1 that may have affected the lack of significant impact of LIPI score on PFS. Another limitation includes the small sample size of the cohort, which did not allow drawing a solid conclusion according to LIPI score and the use of corticosteroid. Moreover, our study did not include geriatric variables. Despite these limitations, we believe that our study revealed for the first time how a simple and accessible worldwide score such as the LIPI score may provide additional information to stratify the prognosis of older patients. Further studies should be performed combining LIPI and geriatric variables, encouraging the use of a comprehensive geriatric evaluation (CGA) in daily routine practice, as well as in clinical and translational research on cancer immunotherapy in older patients, with the aim of enabling them to benefit from modern treatments without impairing their quality of life.

## 5. Conclusions

The LIPI score, based on dNLR and LDH at baseline, was associated with OS in older patients with advanced solid tumors treated with ICBs. It is a simple and accessible worldwide tool that could serve as a prognostic factor in the context of ICB treatment in the older population. LIPI should be prospectively studied in clinical trials.

## Figures and Tables

**Figure 1 cancers-14-05078-f001:**
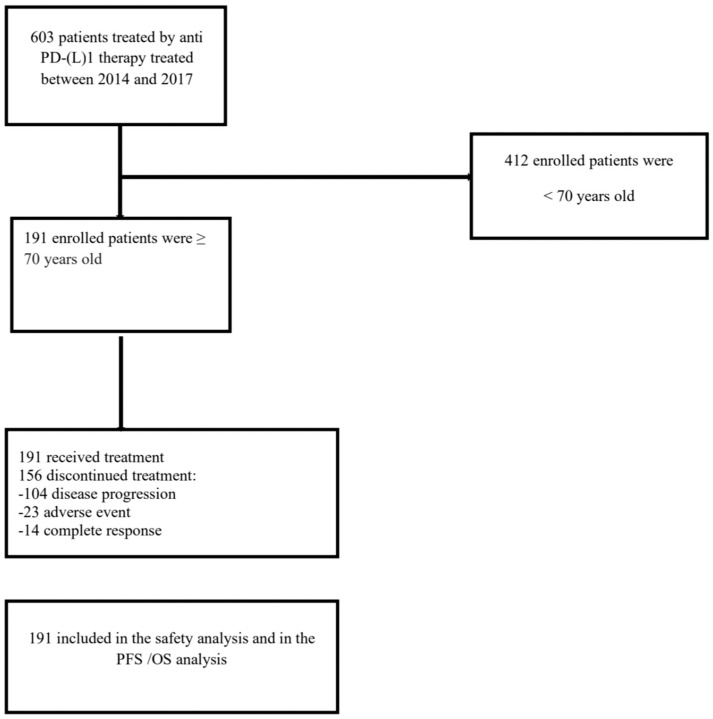
Study flowchart. CONSORT diagram of patients treated with an antiPD-(L)1 monotherapy between 2014 and 2017 in real-life situation, i.e., following marketing authorization, part of an early access program for unlicensed indications or upon compassionate use at Gustave Roussy.

**Figure 2 cancers-14-05078-f002:**
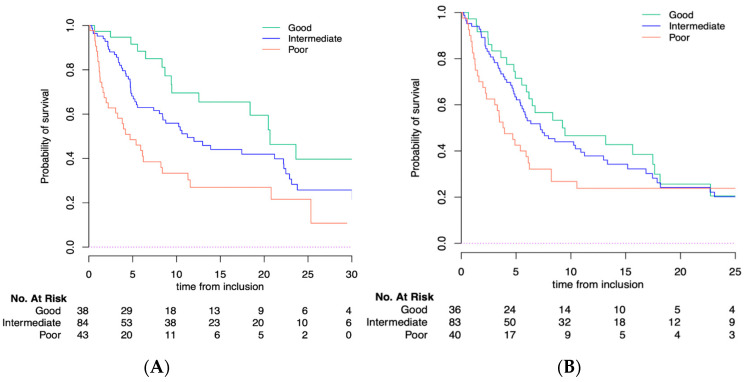
(**A**) Overall survival (OS) and (**B**) progression-free survival (PFS) according to LIPI score in older population.

**Figure 3 cancers-14-05078-f003:**
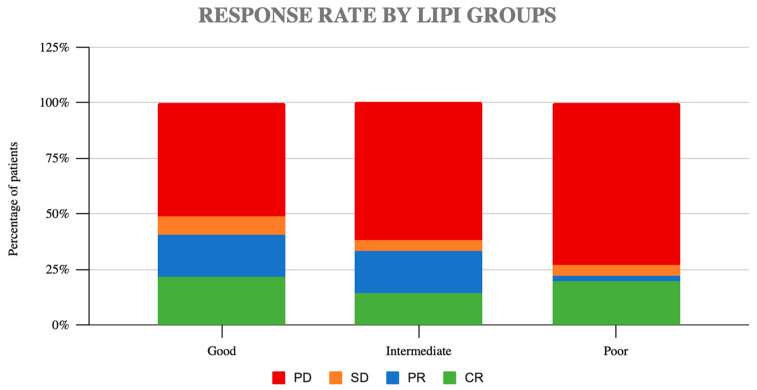
Response rates according to LIPI groups (PD: progression disease, SD: stable disease, PR: partial response, CR: complete response).

**Table 1 cancers-14-05078-t001:** Baseline characteristics of the study population and according to LIPI score (NSCLC: non-small-cell lung cancer; SCLC: small-cell lung cancer; HNSCC: head and neck squamous cell carcinoma; PD-(L)1: programmed death (ligand)1).

	Overall Population, *n* = 191 (100%)	LIPI Good *n* = 38 (23%)	Intermediate LIPI *n* = 84 (51%)	Poor LIPI *n* = 43 (26%)
Age				
median, range	77 (70–93)	78 (70–91)	78 (70–93)	76 (70–89)
Gender				
Female	74 (39%)	20 (52%)	28 (33%)	16 (37%)
Male	117 (61%)	18 (47%)	56 (67%)	27 (62%)
Cancer type				
Melanoma	127 (66.5%)	31(81.5%)	58 (69%)	31(72%)
Merkel cell	2 (1%)	0 (0%)	2 (2.3%)	0 (0%)
NSCLC	29 (15%)	4 (10.5%)	10 (11.9%)	7 (16.2%)
SCLC	22 (11,5%)	0 (0%)	10 (11.9%)	3 (6.9%)
Urothelial	7 (3.6%)	2 (5%)	1 (1%)	2 (4.6%)
HNSCC	1 (0.5%)	1 (2.6%)	0 (0%)	0 (0%)
Main sites of metastasis				
Skin	37 (19.3%)	7 (18.4%)	19 (22.6%)	7 (16.2%)
Lymph nodes	84 (43.9%)	14 (36.8%)	38 (45.2%)	25 (58.1%)
Lung	71 (37%)	11 (28.9%)	28 (33.3%)	23 (53.4%)
Liver	37 (19.3%)	4 (15.5%)	14 (16.6%)	12 (27.9%)
Bones	49 (25.6%)	8 (21%)	18 (21.4%)	12 (27.9%)
Adrenal glands	20 (10.4%)	3 (7.8%)	8 (9.5%)	3 (6.9%)
Kidney	6 (3%)	0 (0%)	2 (2.3%)	1 (2.3%)
Spleen	8 (4%)	4 (10.5%)	3 (3.5%)	1 (2.3%)
Gastrointestinal	11 (5.7%)	3 (7.8%)	3 (3.5%)	3 (6.9%)
Brain	37 (19.3%)	2 (5.2%)	17 (20.2%)	14 (32%)
Thyroid	2 (1%)	0 (0%)	2 (2.3%)	0 (0%)
Pancreas	1 (0.5%)	0 (0%)	1 (1%)	0 (0%)
Performance status ECOG				
0–1	160 (84.2%)	38 (100%)	73 (88%)	32 (74%)
≥2	30 (15.7%)	0(0%)	10 (12%)	11 (25.5%)
Missing	1 (0.5%)	0 (0%)	1 (0.5%)	0 (0%)
Line of treatment				
≤2	170 (89%)	36 (95%)	79 (94%)	33 (77%)
>2	21(11%)	2 (5%)	5 (6%)	10 (23%)
Missing	0	0	0	0
Type of immunotherapy				
Anti PD1	182 (95.2%)	36 (94.7%)	81 (96.4%)	41(95.3%)
Anti PD(L)1	9 (4.7%)	2 (5.2%)	3 (3.5%)	2 (4.6%)
Types of anti PD(L)1				
Avelumab	2 (1%)	0 (0%)	2 (2.3%)	0 (0%)
Atezolizumab	7 (3.6%)	2 (5.2%)	1(1.1%)	2 (4.6%)
Pembrolizumab	114 (59.6%)	29 (76.3%)	50 (59.5%)	28 (65%)
Nivolumab	68 (35.6%)	7(18.4%)	31 (36.9%)	13 (30.2%)
Steroids at baseline				
Dose >20 mg (prednisone equivalent)	25 (13.3%)	2 (5.2%)	11(13.2%)	8 (18.6%)

**Table 2 cancers-14-05078-t002:** Multivariate analysis for progression-free survival (PFS) and overall survival (OS) in the study population.

Multivariate Analysis	PFS			OS		
	*HR*	*95% CI*	*p-Value*	*HR*	*95% CI*	*p-Value*
Gender						
Male	1.201	0.78–1.83	*0.401*	1.392	0.87–2.22	*0.164*
Histology						
NSCLC	1.81	1.08–3.04	*0.05*	1.70	0.97–2.95	*0.06*
Urothelial	2.27	0.66–7.77	*0.05*	2.47	0.62–9.77	*0.19*
Other	3.47	0.81–14.75	*0.05*	4.52	0.58–34.71	*0.14*
Main sites of metastasis						
Lung	1.09	0.72–1.65	*0.66*	1.174	0.75–1.83	*0.48*
Liver	1.50	0.90–2.52	*0.11*	1.55	0.90–2.66	*0.10*
Bone	1.23	0.74–2.05	*0.40*	1.12	0.65–1.91	*0.66*
Adrenal glands	1.90	1.01–3.54	*0.04*	2.64	1.40–4.95	*0.003*
Brain	1.29	0.78–2.11	*0.31*	1.18	0.70–1.99	*0.52*
Immunotherapy line						
>Second line	0.967	0.48–1.93	*0.92*	0.549	0.26–1.16	*0.116*
Concomitant Steroids dose prednisone equivalent						
	2.39	1.32–4.32	*0.004*	2.637	1.44–4.80	*0.002*
PS						
≥2	2.012	1.05–3.84	*0.035*	1.728	0.87–3.41	*0.115*
Albumin						
Low	1.652	0.95–2.86	*0.073*	2.394	1.35–4.22	*0.003*
LIPI score						
Intermediate	0.865	0.52–1.42	*0.36*	1.391	0.77–2.50	*0.008*
Poor	1.224	0.66–2.24		2.77	1.37–5.59	

**Table 3 cancers-14-05078-t003:** Type of irAEs according to the LIPI group. CTCAE, Common Terminology Criteria for Adverse Events).

	Overall Population *n* = 191 (100%)	LIPI Good *n* = 38 (23%)	Intermediate LIPI *n* = 84 (51%)	Poor LIPI *n* = 43 (26%)
irAES	63 (32.9%)	17 (44%)	26 (30.9%)	13 (30.2%)
Median Grade CTCAE	2.43	2.47	2.42	2.46
Types irAEs				
Skin	29 (15%)	5 (13%)	14 (16%)	6 (14%)
Lung	5 (2.6%)	1 (2.6%)	4 (4.7%)	0 (0%)
Liver	6 (3%)	3 (7.8%)	3 (3.5%)	0 (0%)
GI	6 (3%)	2 (5.2%)	1 (1%)	1 (2%)
Thyroid	17 (8.9%)	5 (13%)	8 (9%)	2 (4.6%)
Pancreas	3 (1.5%)	2 (5.2%)	1 (1%)	0 (0%)

## Data Availability

The data presented in this study are available on request from the corresponding author. The data are not publicly available due to ethical reasons.

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
