# Peer review of "Predicting Immunotherapy Outcomes in Older Patients with Solid Tumors Using the LIPI Score"

_cancers, 2022, doi:10.3390/cancers14205078_

Round 1

Reviewer 1 Report (Previous Reviewer 3)

The authors responded to all the comments raised by the reviewers, however unfortunately, they could not provide sufficient data related to the comments to strengthen their hypothesis mainly due to the lack of the data or samples.

Author Response

We thank the reviewer for this comment. Unfortunately, we could not add required data related to the comment because of the lack of the data or the sample, as the reviewer rightly said.

Reviewer 2 Report (Previous Reviewer 1)

The manuscript has been improved. I have no further comments and suggestions.

Author Response

We thank the reviewer for this comment.

Reviewer 3 Report (New Reviewer)

The Authors have showed a very interesting paper on a hot topic in medical oncology, which is the impact of LIPI on clinical outcomes in elderly patients treated with immune checkpoint inhibitors.

The study is well designed and easy to read and to follow. I have only minor comments:

- There are editing from likely previous draft (in red) that should be eliminated.

- Table 1 and 2 are redundant, therefore I suggest to maintain only one.

- I suggest to change "older" in "elderly" patients.

Author Response

We thank the reviewer for this comment.

-We have eliminated table 1 and maintained only table 2.

-We prefer the term older patients instead of elderly patients according to anti-ageism campaigns that suggest to avoid terms as elderly, aged and senior that could falsely present older people as frail, immobile and burdensome.

This manuscript is a resubmission of an earlier submission. The following is a list of the peer review reports and author responses from that submission.

Round 1

Reviewer 1 Report

In this manuscript authors use the “Lung Immune Prognostic Index” (LIPI-Score) to monitor response to therapy and survival in older cancer patients undergoing check point blockade (ICI) Mono-therapy. Authors describe LIPI-Score as a simple score which shows an association to overall survival but not to progression-free survival.
The paper is generally well written. However, there are several issues which should be addressed by the authors.

Major points

  1. Lipi-Score combines dNLR and LDH. The authors have to show, that the combination of the two biomarkers dNLR and LDH is better than using single markers in the special patient group. In receiver operating characteristic (ROC) curve analysis, the area under the curve (AUC) is the most widely used index of diagnostic accuracy for the assessment of the utility of a biomarker. The authors have to include more biomarker indices.
  2. The authors have to better explain the cut-off values used for dNLR and LIPI-Score in the special patient group investigated. Is the cut-off really the same for younger and older patients? How was the cut-off verified in the study? What was the “ULN for LDH”? 
  3. The authors have to compare the “Performance status ECOG” with the Lipi-Score in ROC analysis. If the “LIPI-poor” group was most likely associated with “poor ECOG”, what is the advantage of using LIPI-Score instead of ECOG?
  4. The authors do not show whether the 191 patients ≥70years really have a different outcome in comparison to the 412 patients<70years. Is there a difference between dNLR and LDH and LIPI-Score with respect to the age?
  5. Is there a difference between dNLR and LDH and LIPI-Score in the different tumor histologies? Are there differences for these biomarkers for male and female patients?
  6. The number of lymphocytes decreases with age. Why do the authors use dNLR (with monocytes and lymphocytes in the denominator) and not neutrophil/lymphocyte ratio? Could you show that dNLR is better than NLR in older patients?
  7. Instead of showing all main sites of metastasis in Table 1, the authors should show comorbidity data, such as cardiopulmonary disease, hypertension, diabetes mellitus and dyslipidemia.
  8. Several patients had steroid therapy at baseline. Please show the impact of this therapy on dNLR and LIPI-Score.
  9. Meanwhile ICI monotherapy is not the standard therapy. Could the authors show data or discuss the point, whether combination of ICI with chemotherapy and radiotherapy will have an impact on the LIPI-Score as biomarker.
  10. In Table 3 multiple testing was used. The authors have to state whether Bonferroni correction or similar methods were included for α-failure correction.

Minor points

  1. Line 227: You cite 21 as an example, that high dNLR has been associated with shorter survival in patients with various tumor types, however, 21 is a reference for metastatic lung cancer
  2. Line 209-210: reference 14 is not in brackets
  3. What is CTCAE in Table 4?

Reviewer 2 Report

The authors conducted an important clinical study to understand the prognostic value of the Lung Immune Prognostic Index in senior patients’ treatment with immune checkpoint blockers. The study is based on a large patient number, will be useful in identifying patients who will not benefit from ICB treatment, and the manuscript was well written and easy to follow. Therefore I would suggest publishing this paper in Cancers.

Minor issues:

  1. The text in figure 1 and figure 2 is very small.
  2. The text font in different tables is different.

Author Response

  1. The text in figure 1 and figure 2 is very small.

We thanks the author for this remark. We augmented the text size in Figure 1 and 2. 

  1. The text font in different tables is different

Thanks for this comment. We have made the text font in different tables uniform. 

Reviewer 3 Report

In this study, the authors evaluated the association between LIPI score (combination of dNLR and LDH) and outcomes (PFS, OS, and response rate) of immune checkpoint inhibitors in old patients (70 years old or older) with solid tumors (mainly melanomas and NSCLCs). It would be useful if we can select patients who would have benefit from ICI by this simple biomarker, however, the reviewer raises several major and minor comments.

1. As the authors described in the Discussion, there is no control cohort (older patients who did not receive ICI therapy), therefore, it is unclear if LIPI score is a predictive marker for ICI therapy or LIPI score is associated with patients' outcomes regardless of treatment. It is possible that poor LIPI people will have shorter OS even in general population without cancer.

2. As shown in Figure 3, almost 20% of patients experienced complete response in patients with Poor LIPI score (this is actually higher than Intermediate group). Therefore, it would be inappropriate not to use ICI therapy for all of the Poor LIPI score patients.

3. Other biomarkers, at least PD-L1 expression level, should be added in the analysis.

4. Causes of deaths (cancer-related or non cancer-related) should be included in the analysis. It is possible that there were many non cancer-related deaths in Poor LIPI score group.

5. Were all patients treated with ICI monotherapy? This should be clearly indicated in the Abstract and in the Methods. If so, please add analyses for cohorts who received ICI combo to see if LIPI score was also associated with outcomes. If this cohort include patients who received ICI combo, such data should be added in the multivariate analysis.
